# Broadening the Taxonomic Breadth of Organisms in the Bio-Inspired Design Process

**DOI:** 10.3390/biomimetics8010048

**Published:** 2023-01-23

**Authors:** Amanda K. Hund, Elizabeth Stretch, Dimitri Smirnoff, Gillian H. Roehrig, Emilie C. Snell-Rood

**Affiliations:** 1Department of Ecology, Evolution and Behavior, University of Minnesota, Twin Cities, MN 55108, USA; 2Department of Biology, Carleton College, Northfield, MN 55057, USA; 3Department of Curriculum and Instruction, University of Minnesota, Twin Cities, MN 55455, USA

**Keywords:** divergent thinking, biological models, brainstorming, problem-based learning

## Abstract

(1) Generating a range of biological analogies is a key part of the bio-inspired design process. In this research, we drew on the creativity literature to test methods for increasing the diversity of these ideas. We considered the role of the problem type, the role of individual expertise (versus learning from others), and the effect of two interventions designed to increase creativity—going outside and exploring different evolutionary and ecological “idea spaces” using online tools. (2) We tested these ideas with problem-based brainstorming assignments from a 180-person online course in animal behavior. (3) Student brainstorming was generally drawn to mammals, and the breadth of ideas was affected more by the assigned problem than by practice over time. Individual biological expertise had a small but significant effect on the taxonomic breadth of ideas, but interactions with team members did not. When students were directed to consider other ecosystems and branches of the tree of life, they increased the taxonomic diversity of biological models. In contrast, going outside resulted in a significant decrease in the diversity of ideas. (4) We offer a range of recommendations to increase the breadth of biological models generated in the bio-inspired design process.

## 1. Introduction

Looking to biological adaptations can be a powerful method to inspire novel solutions to a range of challenges [1,2,3,4]. For instance, the structure of butterfly wings has provided new methods for reducing glare on screens [5,6], and studies of gecko feet have inspired the development of novel robotic graspers and reversible adhesives [7,8]. However, biomimetic research tends to represent the tip of the iceberg of biological diversity [9,10,11,12]. The species that are often the focus of bio-inspiration (e.g., geckos, butterflies) are overrepresented in biomimetic research as “biological models” relative to their taxonomic representation [12,13]. While these species have unique traits that have inspired a number of applications, adaptations in less frequently considered species can be just as valuable in bio-inspired applications, such as steerable needles based on parasitoid wasp ovipositors [14]. Even within taxonomic groups, there tends to be domination by particular biological model species. For instance, the blue morpho butterfly is more commonly studied for structural color applications (e.g., [15,16]) than other butterfly species with different mechanisms of manipulating light reflection [17,18]. The taxonomically restricted search space of biomimetics is not inherently a bad thing: we need to focus in depth on particular biological models to understand them enough to abstract design principles and translate them to new technologies. However, we are not realizing the full potential of bio-inspired design until we can more effectively search the entire tree of life for ideas during the initial idea exploration stage [9].

In bio-inspired design, there is an incentive to increase the breadth of total ideas but also the diversity of ideas with respect to evolutionary relatedness, such as that represented by taxonomic diversity. A major limitation to copying biological traits is that natural selection is constrained by available genetic variation and past evolutionary history [19,20], which means that biology is often imperfect from a design perspective and that closely related species will often have similar limitations [21,22]. However, engineers and designers are not limited by the same processes and can mix and match mechanisms from different biological models. Because the independent origins of a trait or function across species are often underlain by different mechanisms (i.e., distantly related species often solve the same problem in different ways), exploring a range of examples across different parts of the tree of life means that a biodesigner is more likely to see different traits and mechanisms [23,24,25] and come up with better solutions.

To search a broader range of biological models for inspiration, we might consider interventions at the initial stages of the bio-inspired design process. In many ways, broadening this idea-space is analogous to a creativity exercise, where we seek to brainstorm a broader range of ideas, also termed “divergent thinking” [26,27]. Divergent thinking is one of many factors that can lead to original, creative, and innovative ideas [28]. Several factors may limit such divergent thinking in a biomimetic process. First, prior experience tends to bias us toward biological models with which we are more familiar [29,30], as we often have the knowledge of these organisms to make the analogy bridge between biology and the focal problem. Databases such as *AskNature* can help those new to biology discover different organisms [31,32], but studies have shown that this is still a narrower range than that generated by biologists with more familiarity with biodiversity [11,33]. Tendencies toward familiar organisms further restrict our idea generation, similar to other cognitive biases (e.g., confirmation bias [34,35]). For instance, humans tend to be particularly drawn to mammals and “charismatic megafauna” [36,37], which may limit biomimetic analogies to large animals and mammals [12] more than other taxonomic groups, even when other species are better models or present novel solutions to a problem.

The creativity literature offers several solutions for encouraging more divergent thinking during the bio-inspired design process. In this research, we focus on three general ideas for increasing the breadth of biological ideas generated during the brainstorming process. First, most people get better at creativity tasks with practice [38,39,40]. Giving biomimetic researchers opportunities to develop divergent thinking skills related to biological analogies is likely important: practice generally improves performance and alters neural structures [41,42]. Alternatively, it is possible that the focal problem, communicated in a given brainstorming prompt, may override or obscure the effect of practice (the role of **prompt versus practice**).

Second, we focus on the role of individual expertise versus that of collaborators or team members (the role of **expertise**). The literature suggests that interactions in diverse teams of collaborators can encourage divergent thinking [43]. In particular, diversity with respect to knowledge and expertise can significantly improve team creativity [44,45,46]. With respect to biomimetics, we might expect the key piece of collaborator diversity to be complementary expertise across different taxonomic groups. In other words, learning from others with different biological expertise should broaden the range of ideas generated in the bio-inspired design process. At the same time, an individual’s own expertise may be just as important as what they are learning from their peers.

Third, we tested the efficacy of two interventions that we designed to increase the range of ideas generated (the role of **interventions**). Research has suggested that time outdoors in nature may improve creative thinking [47,48]. We might expect this to be especially true with respect to ideas related to biological models. In a nature “wandering” exercise, students may notice new biological models they had not thought of and relate them to the problem at hand (the **outdoor** intervention). We also designed an intervention to explicitly incorporate taxonomic diversity into brainstorming related to bio-inspired design. We reasoned that the breadth of biological models generated would be even more powerful if they were distantly related and thus more likely to use different mechanisms to get at the same function. We developed a novel intervention, where students are somewhat randomly pushed into different parts of the tree of life and different ecosystems across the Earth (the **idea-space** intervention).

In this research, we used a classroom setting to test why people vary in their divergent thinking during the bio-inspired design process. We used an Animal Behavior course where the typical topics of such a course (e.g., sexual selection, cooperation, biomechanics, habitat choice) were “flipped” through a bio-inspired lens (e.g., [49]). The course was fully online during the COVID pandemic (spring 2021) and consisted of 180 students. For each week, students were assigned a brainstorming problem, where they had to brainstorm a range of biological models that would be ideal to study for inspiration in solving a given problem (Table 1). This range of models was used as a starting point for the content of the course: after discussing relevant basic research, the course returned to the problem and considered what (if anything) these models might offer for ideas for problem solving. We used a series of brainstorming activities to test three sets of ideas as to why divergent thinking varies: (1) prompt versus practice, (2) individual versus team member expertise, (3) the role of two interventions—going outside and using online tools to explore novel parts of the tree of life. We predicted that each of these interventions would increase the range of biological models generated during brainstorming with respect to both the total number of ideas and the taxonomic diversity of ideas.

## 2. Methods

### 2.1. Course Details

In this research, we studied the brainstorming exercises of 180 students enrolled in a course on Animal Behavior. Students were primarily upper-level biology students, including majors in Neuroscience (20%), Genetics/Cell Biology (13%), and General Biology (44%). By credits, 86% of students were seniors, 13% were juniors, and the remainder were sophomores or non-degree-seeking students. Most students in the course were on a pre-health career track. During the spring semester of 2021, this class was online due to the COVID-19 pandemic.

### 2.2. Assignment Design

Students had brainstorming assignments due every week (on Thursday, for discussion the following Monday). Prompts across weeks were similar in that students were assigned a focal problem, which was often briefly discussed in class (Table 1), and then they were asked to brainstorm a list of animal systems they would study for inspiration. For two of the prompts, students could choose one of two options within a category (Mental Health: “Grief” or “Anxiety”; Healthy Living: “Nutrition” or “Movement”). Additional portions of the assignment varied by week depending on the interventions we were testing, and other moving parts of the course (e.g., literature research or learning about manuscript writing for this writing-intensive course). The complete texts of the assignments and the list of assignments and instructions can be found in Appendix A. In sum, there were 11 pre-class brainstorming assignments, but students only had to complete 10 for 15% of their total grade. The first of these assignments (our baseline assignment) was not due until the second Thursday of the course, so students had plenty of time to learn about the assignment.

For our analyses, we chose a subset of assignments to analyze in part due to the time cost of scoring 180 submissions for each assignment. We focused on assignments that allowed us to test three sets of ideas as to why divergent thinking varies: (1) prompt versus practice, (2) individual versus team member expertise, (3) the role of two interventions—going outside and using online tools to explore novel parts of the tree of life. We chose the first brainstorming assignment (“force”) as a baseline. The final two assignments (“cooperation” and “communication”) were the focus of the outdoor and the idea-space interventions (see more details below). We chose two additional assignments partway through the course to test the problem vs. practice and expertise hypotheses. The “mental health” assignment was chosen, as it occurred about ⅓ of the way through the course, after many breakout room discussions within student teams that varied in biological expertise (see below). The “healthy choices” assignment was chosen at a later time point (about ⅔ of the way through the course). Both “mental health” and “healthy choices” had two options for students to choose from, allowing us to further test the influence of the prompt separately from practice (time in the course). This collection of five assignments for 180 students (for a total of 775 scored student responses from 178 unique students) allowed us to test the three focal hypotheses, teasing apart the effects of prompt vs. practice, individual and team expertise, and specific interventions (see below).

### 2.3. Prior Knowledge and Learning from Team Members

To test hypotheses about students’ prior knowledge of biodiversity and the taxonomic expertise of team members, we gathered data before the course on their familiarity with eleven taxonomic groups. Using a survey, we asked students to estimate their familiarity with mammals, birds, reptiles/amphibians, fish, insects, arachnids and relatives, crustaceans, mollusks, “all the worms,” cnidarians, and sponges. For each taxonomic group, students assessed their knowledge on a scale of 1 to 10, corresponding to “know little” to “know lots!”, respectively. In total, 177 students filled out the survey in 2021, although given that some of these dropped the course, in the end, we had 165 (of 180) students with data on taxonomic expertise (distribution of responses in [50]). Two metrics were taken from the biodiversity knowledge surveys: an individual’s “total expertise,” that is, a summary of scored expertise across the eleven taxa, and a measure of their diversity of knowledge using a commonly used measure of diversity from the ecology literature—the Shannon index (Sum(pi(LNpi)) across all taxonomic groups), which adjusts for the evenness of knowledge across taxonomic groups, with higher values representing more consistent knowledge across taxonomic groups.

To test the role of individual expertise versus what students were learning from their peers, we compared the effects of their individual expertise versus that of team members with whom they interacted *after* a brainstorming assignment (i.e., group interactions were expected to affect individual brainstorming exercises later in the semester after multiple assignment discussions). Student teams were constructed prior to the first day of the course in a semi-systematic, semi-randomized way. Expertise was first sorted within each taxonomic group, and then student teams were assigned (up to 30 total teams to distribute “experts” in different taxonomic groups across student teams). Otherwise, sorting into student teams was performed blind to student identity. We were sure to account for variation across teams in students missing taxonomic score data (these individuals were spread evenly across student teams 17–30). Our method of constructing teams seemed to work well, as there were no significant differences in individual total expertise scores across the 30 student teams (total expertise: F_29,136_ = 0.47, *p* = 0.98), nor was there a difference in Shannon diversity scores across student teams (F_29,136_ = 0.71, *p* = 0.86).

Student teams (breakout groups) were the same throughout the semester. Students went into these teams at least once per class, sometimes 2–3 times in a class session (of 75 min). During the first breakout session on the first day of the course, students were asked to complete a team norming exercise where they shared some information about their backgrounds and worked out guidelines for interactions over Zoom. Other team activities included sharing ideas from pre-class assignments, designing a composite “sensory robot” (for a team prize), and completing statistical problems together.

In our analyses, we focused on an individual’s incoming expertise scores and how they influenced their response to the first individual brainstorming assignment (“force”), which came at the beginning of the course before team interactions. We were also interested in whether students could learn from team members with different taxonomic expertise, which was predicted to broaden the diversity of ideas when completing individual brainstorming assignments later in the semester. To do this, we looked at an individual’s expertise scores relative to their team members (student score minus the average of the other team members) to see how this influenced responses on later assignments that occurred after team interactions (“grief”, “anxiety”, “movement”, “nutrition”). Thus, positive values indicated that an individual self-reported higher expertise than their team members, while negative values indicated that an individual self-reported lower expertise than their team members.

### 2.4. Outdoor and Idea-Space Interventions

We designed two interventions and measured the impact on student responses. First, in the idea-space intervention, we were interested in how the explicit consideration of “evolutionary space” and “ecological space” would increase the taxonomic diversity of student ideas (Appendix A). Students were instructed to think about how organisms across different branches of the tree of life evolved traits to perform the same function. They could use various web tools to explore evolutionary space, such as a website that takes the user to a random organism on the tree of life (tol.org) or Wikipedia, by using the taxonomic classifications of different organisms. Students were also encouraged to explore different biomes or geographic regions (phrasing listed in Appendix A). In the outdoor intervention, students conducted a nature-wandering activity, where they walked through an outdoor space and took notes on the animals they saw that might be related to the focal problem for the week (“cooperation” or “communication”). Students were split into two sets by their last name and completed both interventions and topics in a factorial manner to control for the topic and order (Appendix A).

### 2.5. Assessment of Divergent Thinking

We focused on four measures of divergent thinking—or the breadth of biological models generated during brainstorming about different problems (all assignment prompts are in Appendix A). First, we counted the total number of examples generated for the assignment. To measure the taxonomic breadth of the responses, we counted the total number of taxonomic classes represented in this list (e.g., birds, mammals, bony fish, insects). We focused on “class” as the taxonomic level (as opposed to genus or family) because this is the level where we generally learn about and study animal diversity (e.g., diversity classes are taught around “mammalogy,” “ornithology,” “entomology”). To assess mammal bias, we counted the number of examples that were mammals and calculated the “proportion mammals.” Finally, we measured the specificity of the responses as the number of examples that were a specific past taxonomic class; for instance, “bird” was not specific while “sparrow” was.

### 2.6. IRB Approval

This research was conducted under IRB–STUDY00010075 (Approved on 18 June 2020). Students were informed of this creativity study on the first day of the course. On the second day of the course, students watched a brief video prepared by the student investigator that explained the study. Students opted into the study by agreeing to participate in pre- and post-surveys on creativity (not analyzed in the present study). Otherwise, all present research made use of regular classroom activities, which were a component of the students’ grades.

### 2.7. Assessment of Student Engagement

We wanted to ensure that the online format of the class did not interfere with student engagement with the material (and thus components of the study). We compared the results from the 2021 online course to those from the 2018 in-person course. The 2018 course was similar in content but in-person and not taught through a bio-inspired design lens. We chose this year instead of 2019 or 2020 because the instructor was on sabbatical in 2019, and the spring 2020 semester was atypical in terms of methods, student stress, and assessment structure due to COVID.

### 2.8. Statistical Analyses

We ran all statistics and made all figures using R version 4.2.1 (R team 2022). We first focused our analyses on the role of the prompt versus practice using data from all of the topics apart from the two assigned to the two interventions (“force”, “grief”, “anxiety”, “movement”, and “nutrition”). We built general linear mixed models with students as a random effect using the *lme4* package [51]. We checked all models for necessary assumptions, including residual fit and overdispersion. Our response variables included the number of examples (Poisson model), the number of taxonomic classes represented (Poisson model), the proportion of responses that were mammals (Binomial model), and the proportion of responses that were specific (Binomial model). Given that the effect of the prompt topic was significant, we looked at least-squares means to explore the pairwise contrasts (*t*-tests) between different prompts. We could not repeat prompt topics randomly for different students at different times in the course (the prompts were tied to course content) and, thus, could not test for the impact of practice (time in the course) separately from the prompt. However, we could still use the results from our models to assess the effect of practice by looking at the order of the prompts and how this order influenced our different response variables.

Second, to assess the role of individual versus team member expertise in the breadth of ideas, we initially ran analyses for just the first prompt (“force”). We used a Spearman rank correlation test for each of our response variables (breadth of ideas: number of examples, number of taxonomic classes, mammal bias, specificity) to test for the effects of an individual’s taxonomic expertise. Then, we turned to the later prompts (“grief”, “anxiety”, “movement”, and “nutrition”), which came after students had interacted with team members in breakout rooms, sharing their different examples for previous assignments. We used a metric of individual expertise relative to team member expertise and used the non-intervention prompts that occurred after teams had worked together. We used general linear mixed models, as described above, and included student and prompt topic as random effects.

Finally, to see if our different interventions (none, outdoor, or idea-space) influenced the creativity of student responses, we used the full data set and built general linear mixed models for each of our response variables, as described above, and included student and prompt topic as random effects. To illustrate our results, we present violin plots, which are a hybrid of a boxplot and a kernel density plot. Violin plots contain all data points and depict both summary statistics and the density of each variable and are appropriate for data that do not conform to a normal distribution. We also present results with word clouds as a visual representation of the list of responses or taxonomic classes, where word size represents frequency.

## 3. Results

### 3.1. Overview of Assignment Participation

Across 180 students and 11 activities (10 required), students completed 2013 total assignments. These activities were graded for completion, and the average student grade for the overall “weekly class preparation” assignment was 91% (completing 9 out of 11 prompts would be scored as 90%), although 129 (of 180) students received a perfect score of 100% (completed at least 10 out of 11 prompts). To ensure that students were engaged in the class (and thus the assignments) despite the online format, we compared student evaluation data from the present semester to the previous in-person semester. For the online semester, *more* students stated they strongly agreed with the statement “my interest in the subject matter was stimulated by this course” on the end-of-semester evaluations (X2 = 10.61, *p* = 0.01; Appendix A).

### 3.2. Effects of Prompt and Practice

The prompt topic had a significant effect on the number of taxonomic classes represented, with “force” having the most and “grief” and “anxiety” having the least (F = 14.70, *p* < 0.001, pairwise comparisons in Figure 1A). We found a similar pattern for mammal bias (F = 58.15, *p* < 0.001, pairwise comparison in Figure 1B). Figure 2 also illustrates these patterns using word clouds made from student responses and the taxonomic classes represented for the “force” and “grief” topics. The prompt topic did not significantly influence the number of examples given or the specificity of the examples. By looking at our prompt topics in the order they were presented in the course (Figure 1, colors), we found that practice (time in the course) did not appear to influence our measures of divergent thinking in a significant way, at least not in any way stronger than the effect of prompt.

### 3.3. Effects of Individual and Team Member Familiarity with Biodiversity

The amount of total expertise that students brought with them to the course had a small but significant positive influence on the number of taxonomic classes that were represented in their responses to the “force” question (rho = 0.181, *p* = 0.02, R^2^ = 0.012, Figure 3). This was not true when we measured expertise using the Shannon diversity score, which captures the evenness of knowledge across taxonomic groups. We then considered the breadth of biological models brainstormed by individuals later in the course, after they had interacted with team members. Individual expertise had no impact on the number of examples, specificity, or mammal bias (for total expertise score or Shannon diversity score), nor did student expertise relative to their group (for grief, anxiety, movement, and nutrition assignment prompts).

### 3.4. Idea-Space and Outdoor Interventions

Our different interventions (outdoor and idea-space) significantly influenced the number of examples that students gave (F = 39.33, *p* < 0.001, pairwise comparisons in Figure 4A) relative to the exercises that came earlier in the course (labeled “none” in Figure 4). In particular, students in our idea-space intervention gave the greatest number of examples, and students in the outdoor intervention gave the least number of examples. We found a similar pattern for the number of taxonomic classes represented (F = 7.99, *p* = 0.003, pairwise comparisons in Figure 4B). The interventions also influenced how specific the examples were (F = 115.33, *p* < 0.001, pairwise comparisons in Figure 4C), but in this case, responses in the initial assignments (force, mental health, healthy choices) were the most specific, followed by the idea-space intervention, with the outdoor intervention being the least specific. Finally, we found that both the idea-space and outdoor interventions helped reduce the mammal bias in the responses relative to earlier exercises in the course (F = 4.90, *p* = 0.01, pairwise comparisons in Figure 4D). We also illustrate some of these patterns in Figure 5 using word clouds to compare student responses in the idea-space and outdoor interventions to the same question topics, “Communication” and “Cooperation”. We asked students to reflect on their experiences in these two intervention assignments, and a representative subset of their reflections can be seen in Appendix A.

## 4. Discussion

### 4.1. Findings

Generating a range of ideas early in the problem-solving process is an important driver of original, creative, and innovative ideas [28]. In this research, we sought to increase the breadth of analogous biological models generated during the bio-inspired design process. We have much to learn from the adaptations of over ten million species on Earth, but exploring biodiversity for ideas can be challenging, as we are limited by our own biases and knowledge. The taxonomic diversity of ideas is often important in bio-inspired design because evolutionary constraints can result in traits that are imperfect from an engineering design perspective. A way around this is to look to how different organisms solve the same problem in different ways and mix and match strategies in one’s own application. Thus, we were interested in how to push students away from inherent biases toward familiar mammals to consider a range of other models. Our results suggest that we are indeed drawn to mammals and “charismatic megafauna” in our brainstorming, especially for certain topics. However, we found evidence that some interventions could increase the range of biological models generated.

In our consideration of the prompt versus practice, we found no support for the hypothesis that students improve in their brainstorming ability with time, across assignments [38,39]. Instead, the primary driver of divergent thinking across initial assignments seemed to be the assignment prompt. In particular, prompts about problems that were more related to cognitive decisions and mental states resulted in lists with the lowest taxonomic diversity and a much greater mammal bias (Figure 1 and Figure 2). For instance, “anxiety” had huge skews toward mammals, despite the prevalence of fear responses across animals. Even “nutrition” (i.e., foraging decisions) had a high mammal bias, despite foraging behavior being nearly universal across animals. In contrast, the “force” prompt, the first assignment of the semester, resulted in the highest diversity of responses. The bias toward charismatic species and mammals parallels human biases in other areas, such as conservation funding [36,37]. Our finding, that the breadth of brainstorming did not improve over time, is consistent with other studies critiquing the idea of creativity training, which instead suggests that the observed effects of practice are more consistent with learning to do better with particular brainstorming prompts [52]. In the case of ideas for bio-inspired design, it is possible that improvement with time comes more from learning the natural history of many species, rather than practicing with the activity per se. To that end, encouraging students to take biodiversity survey courses (e.g., ornithology, entomology) in parallel with a bio-inspired design course may be a more effective intervention.

We found moderate support for the idea that prior biological knowledge is related to the range of biological models generated. There was a small but significant relationship between an individual’s self-ranked familiarity with a range of animal groups and the taxonomic diversity of models brainstormed in their first assignment (in terms of the number of taxonomic classes of animals). This finding parallels findings that when engineers collaborate with biologists with broad expertise on biodiversity, their lists of biological analogies are more diverse [11,33]. However, in our results, the correlation coefficient between expertise and the diversity of ideas was modest (~0.1), suggesting many other factors are at play. As we asked students to self-assess their knowledge, it is likely that our metrics are not a very accurate indicator of biological knowledge; indeed, students with some familiarity with a taxonomic group may be aware of all of the species they do not know (under-assessing knowledge), just as much as some students may over-assess their knowledge. A quiz on biological knowledge would likely be a better indicator for future tests of this idea.

We found no support for the idea that individuals in student teams with members with rich biodiversity expertise benefitted in terms of the breadth of their ideas later in the semester. This is contrary to expectations from the literature suggesting that the diversity of deep expertise within a group of collaborators leads to more creative thinking and idea generation ([44,45,46], although these studies tend to focus on professionals instead of students). These results are suggestive of observations that the quality of team interactions is key, as creativity can be stifled by conflict and stress [53]. Diversity within teams is only beneficial if there are mechanisms in place to promote meaningful interactions and a culture of respect, equality, and inclusion [46,54,55,56]. To this end, the online nature of this course imposed serious constraints. While students were in the same teams throughout the semester, these were online breakout groups on Zoom, limiting the ability to build familiarity over time, which is key for group dynamics around creativity [57,58]. Students started the semester with a team norming exercise to discuss pet peeves in group discussions and how they wanted to run their discussions, but there was no oversight of this within these 30 teams. In-person interactions where the instructor(s) could wander between teams could help foster more engagement and meaningful team interactions. It is also possible that team interactions simply had limited effects on these individual assignments, and working together to generate team lists may have been more effective [59].

We tested the effectiveness of two interventions to increase the breadth of divergent thinking. We found strong support for our “idea-space” intervention, which used online tools to explicitly push students into new parts of the tree of life or new biomes or geographic areas for ideas. This intervention increased the total number of ideas and the taxonomic breadth of these ideas relative to all other exercises in the course (even though in many of the prior exercises, students were also using Google to help generate their lists). While this intervention was statistically effective, comments from students stressed several challenges with this intervention (Appendix A). First, the assignment itself was complex, suggesting future iterations should break it into smaller pieces, for instance, by focusing on evolutionary space one week and ecological space another week. Second, it was clear that students were limited by their own knowledge of the species that would pop up in their search. A name and an image of an animal do not capture the natural history of the biological model that is necessary to draw an analogy to the assigned problem. Third, in efforts to sample other parts of the evolutionary tree, students kept getting pushed into the hyperdiverse parts of the tree, such as insects (or beetles), sometimes to the detriment of exploring other poorly explored branches. Finally, students seemed to like this intervention less, possibly reflecting discomfort with being pushed into areas that were less familiar and where they had less knowledge of biological diversity. However, small amounts of discomfort can be important in learning new things and promoting divergent thinking, but the proper support during such uncomfortable explorations is needed [60].

We were also interested in the role of going outdoors in the creative process. We found no support for our outdoor intervention increasing the breadth of ideas generated, contrary to predictions from the literature [47,48]. Somewhat surprisingly, the range of ideas generated was actually lower than in previous assignments (Figure 4 and Figure 5). However, the students clearly enjoyed this activity and engaged with nature, even in urban environments, suggesting that the exercise still has important value. Ideas generated outdoors tended to converge on a number of common and easily observable models, such as squirrels, humans, and dogs (which are especially observable during April in Minnesota). These findings are consistent with at least one study that found that going outside alone does not necessarily increase creativity [61]; instead, it may need to be coupled with other interventions. Revisions to this assignment might include explicit instruction to break out of a routine when wandering outside, such as “bring a trowel and dig through the dirt and leaf litter” or “use binoculars and a field guide to think about different species you observe while wandering.” In addition, noticing new things is often challenging in familiar environments, suggesting that students should be encouraged to visit habitats separate from their typical daily routine. Indeed, the few students who visited zoos or state parks included unique species relative to other students. This exercise also reflects the constraints of an outdoor activity at the end of winter in Minnesota; encouraging students to repeat this activity at different time points or in different biomes (e.g., on spring break) could increase its effectiveness.

### 4.2. Recommendations and Limitations

This research provides several recommendations for increasing the diversity of ideas generated during the bio-inspired design process. First, biodesigners should be aware of their taxonomic biases that come into a brainstorming session, especially for problems that may be considered more human-centric. They can use online tools such as Wikipedia or the tree-of-life “random page” generator to push them into new parts of the tree of life or different ecosystems to increase the breadth of biological models considered. Second, bio-designers may benefit from cultivating knowledge of the biology of several groups of organisms and learning about the natural history and diversity across species. Individual expertise is important (e.g., Figure 3), but the most common hang-up in searching for ideas is limited knowledge of the biology of a model, as this is key to making the analogy bridge between biology and design. Third, partnerships with biologists with complementary expertise can be incredibly important, but interactions must be structured in a meaningful and respectful way, and brainstorming performed collaboratively is likely to have a bigger impact on these team interactions (e.g., idea pooling [59]). Finally, going outside and getting into biology often spurs engagement with organisms but does not necessarily increase the breadth of biological models generated. We might consider outdoor interventions to increase connections to the natural history of some core, common organisms, which we can then build on through online explorations to increase the breadth of ideas. Alternatively, we could couple outdoor activities with additional interventions that increase exposure to new organisms, whether it is bringing a trowel to dig in the ground or going to a new location or habitat.

While our findings suggest several broad take-homes with respect to divergent thinking in the bio-inspired design process, it is important to note that there may be limitations in extrapolating from our study population. Our approach was constrained by the topic of the course (Animal Behavior) and the typical population that enrolls in this course. The students were all undergraduates studying biology, not designers and engineers. While this no doubt affects the number of biological examples that come to mind for the prompts in this study, it is not clear whether the relative effects observed here would differ for non-biologists. For instance, we might expect that the relative performance of the two interventions would remain the same. However, it is likely that the effect of learning from others could be more important for a population with less prior knowledge of biology. In addition, it is possible that students from backgrounds with more training in creativity would be inherently better at brainstorming tasks once they are armed with tools to gather biological knowledge.

Throughout this work, we have stressed the importance of exploring a range of biological models across the tree of life in the initial stages of a bio-inspired design process. However, it is important to note that in the subsequent steps of a bio-inspired design process, the biodesigner may indeed choose to work with common or standard biological models. There are many reasons why choosing to work with these models can be beneficial. First, we are often biased toward models where we have more background knowledge (familiarity begets more studies and more familiarity). We often need an in-depth understanding of how traits work to abstract principles relevant to design. For instance, within studies of structural color, there is a large bias toward butterflies, despite structural color in many other species (e.g., some plants and beetles). There is now a huge body of literature on how butterfly wing scales function to reflect light in specific ways—necessary knowledge for the translation to something such as screen design—but there are relatively few studies on mechanisms in other species. Second, for some problems, it is possible that some lineages of life are truly more relevant. For instance, models relevant to studying the problem area of “grief” do tend to be larger-brained and social species; however, these are not limited to mammals (parrots, magpies, and ravens also show evidence of grief [62]). Third, for some bio-inspired applications, we may need to constrain ourselves to specific branches of the tree of life to increase the chances of translation. For instance, in many medical applications, we may increase the chance of a successful translation by studying species more closely related to humans. When beginning a bio-inspired design process, it may help to step back and identify whether we want to overcome biases or embrace them.

### 4.3. Future Research

This work highlights several important areas for future research and development with respect to the bio-inspired design process. First, we are missing search tools that promote taxonomic breadth while also integrating the biological and natural history knowledge necessary for making the analogy between a design problem and the biological model of interest. For instance, one has to know something about how the mantis shrimp hunts to know that it is a good model for thinking about force generation [63]. Tools such as *Google Scholar* or *Ask Nature* allow a biodesigner to explore adaptations around form and function but have no integrated ability to “diversify” the list with respect to evolutionary relatedness or biogeography. The ability is possible in terms of machine learning (e.g., an algorithm figuring out the organisms studied in a paper) and phylogenetic databases (e.g., classifications in Wikipedia are generally up to date), but such a search engine has yet to be built (to our knowledge). Second, our interventions and the qualitative responses of students suggested a variety of exercises that would have an even greater impact on divergent thinking in the bio-inspired design process, but future studies would need to assess their effectiveness. For instance, does coupling an outdoor intervention with direction to look through binoculars increase the range of ideas generated, or does the student have to go outside with a field guide or a fellow student who has a deep knowledge of birds? Does exploring the tree of life result in a more creative list than considering different geographic areas? Third, while our assignments focused on the brainstorming list, we did not assess the quality of the analogy between the focal problem and the biological model generated. Adding questions to assignments to assess the link with the concept of function would allow us to make progress here. Finally, how do these brainstorming exercises, which were explored in a classroom setting, translate to actual bio-design processes where biologists, designers, and engineers are collaborating to build something? Future work can test whether these interventions influence whether resulting designs are more innovative, effective, or sustainable when they come from more diverse lists of possible biological models.

### 4.4. Conclusions

In this research, we found strong support for the hypothesis that the explicit consideration of evolutionary and ecological relationships can increase the taxonomic breadth of biological models generated during the brainstorming phase of a bio-inspired design process. An individual’s prior expertise with biological diversity contributed somewhat to their breadth of ideas, but exploring with online tools such as Wikipedia or the “tree of life” can push students into new idea spaces. We found no evidence that going outside per se increased the taxonomic breadth of ideas, but it is possible that combining this activity with more directed activities (e.g., “look under a rock”) would encourage students to move beyond the most obvious and familiar organisms encountered outdoors. Based on these findings, we generated a number of recommendations for idea generation in bio-inspired design and future research.

## Figures and Tables

**Figure 1 biomimetics-08-00048-f001:**
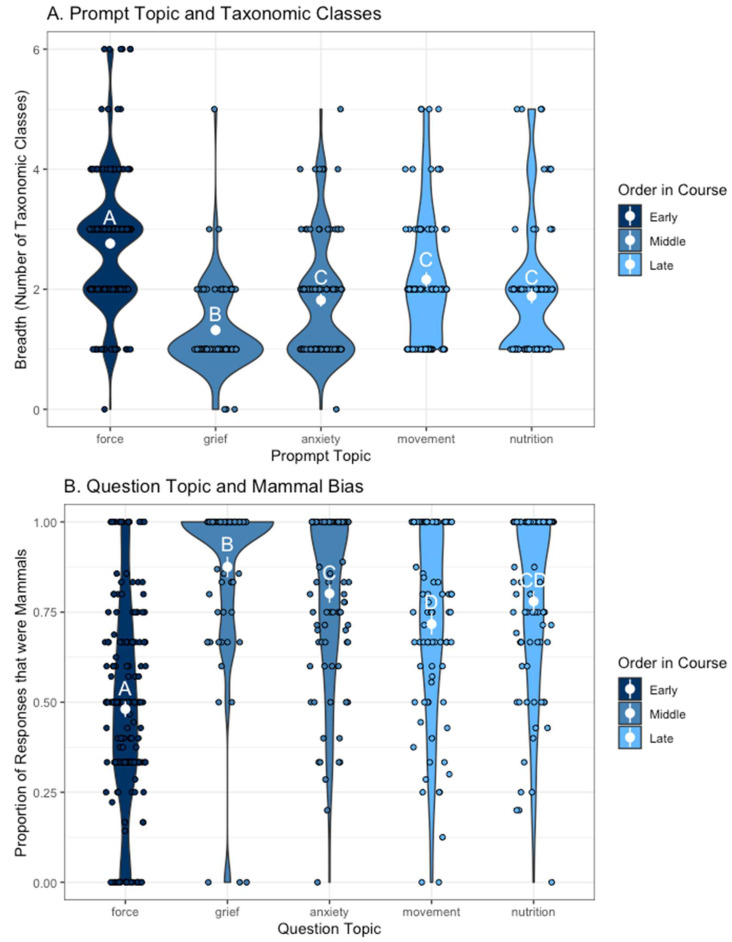
Violin plots illustrating the relationship between the prompt topic and (**A**) the number of taxonomic classes represented (breadth) in the list of examples given by each student and (**B**) the proportion of examples that were mammals (mammal bias). Figures were made with raw data, and analyses presented in the text include student as a random effect (n = 492). “Force” had greater breadth (number of taxonomic classes represented) and less mammal bias (proportion of examples that were mammals) compared to other prompt topics. “Grief” had the least taxonomic breadth and the most mammal bias, while anxiety, movement, and nutrition were intermediate for both these measures of divergent thinking. The temporal order these prompts came in the course is represented by color (e.g., grief and anxiety were assigned the same week) and appeared to have less of an effect on student responses compared to the topic of the prompt. Letters on the graph denote significant differences using pairwise *t*-tests (*p* < 0.05); raw data are shown as jittered points.

**Figure 2 biomimetics-08-00048-f002:**
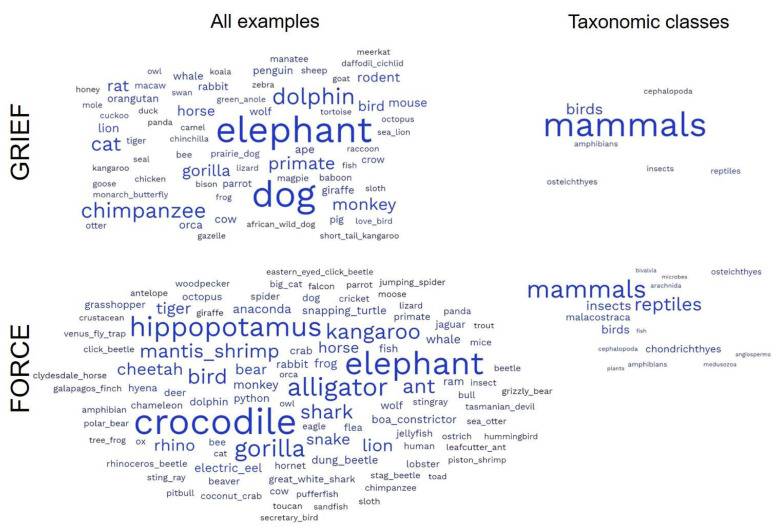
Word clouds for the “grief” and “force” assignments made from all student responses and from the list of taxonomic classes represented in each student’s set of examples. Word size corresponds to frequency across all student responses. Not surprisingly, responses to the “force” prompt had a greater diversity of taxonomic classes represented, and fewer of the responses were mammals (mammal bias) compared to the “grief” prompt. Raw frequency data can be found in [50] under an open access license.

**Figure 3 biomimetics-08-00048-f003:**
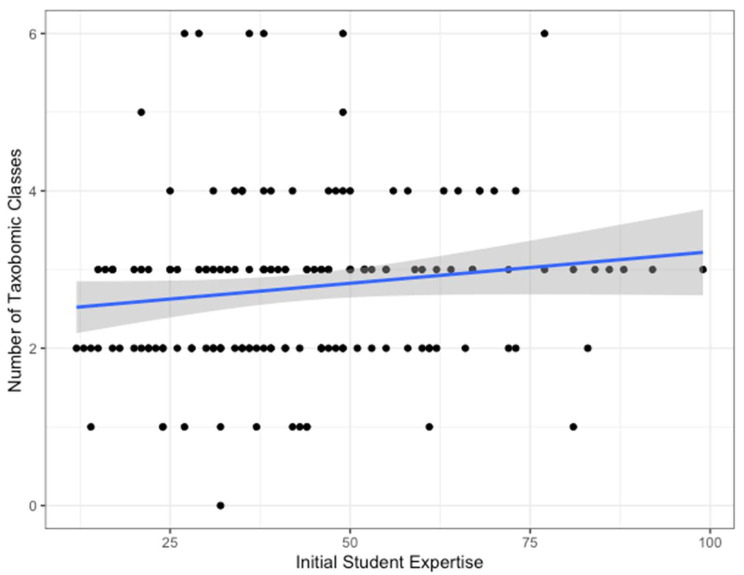
Initial student expertise (total expertise score) had a small but significant effect on the number of taxonomic classes represented in their list of examples given for the “force” assignment. The blue line represents a linear model fit to the data (R^2^ = 0.012) with the standard error in gray. Analyses presented in the text are from a Spearman rank correlation test.

**Figure 4 biomimetics-08-00048-f004:**
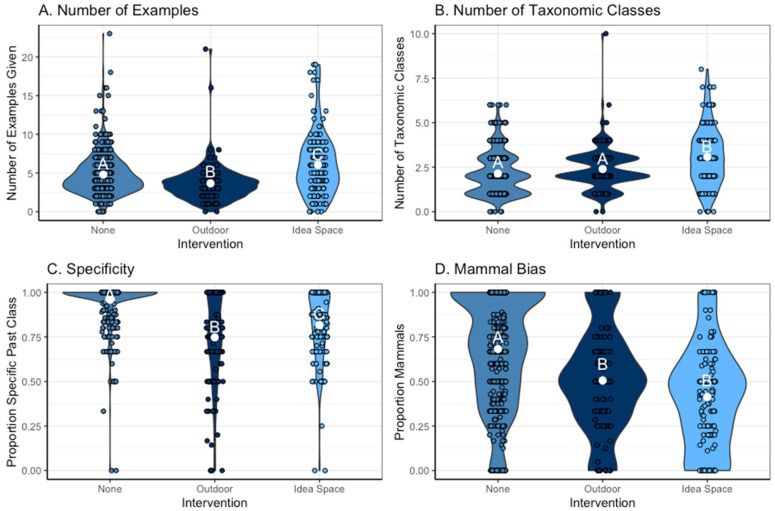
Violin plots visualizing comparisons between the three interventions, none, outdoor, and idea-space, and student responses, including (**A**) the number of examples given, (**B**) the number of taxonomic classes represented (breadth), (**C**) the specificity of the responses (proportion of examples given that were a specific past class), and (**D**) mammal bias (proportion of examples given that were mammals). Figures were made with raw data, and analyses presented in the text include student and question topic as random effects (n = 775). Students in the idea-space intervention generated more responses, with a greater number of taxonomic classes represented, and their responses were more specific and had a lower mammal bias compared to the no-intervention treatment. Letters on the graph denote significant differences using pairwise *t*-tests (*p* < 0.05); raw data are shown as jittered points.

**Figure 5 biomimetics-08-00048-f005:**
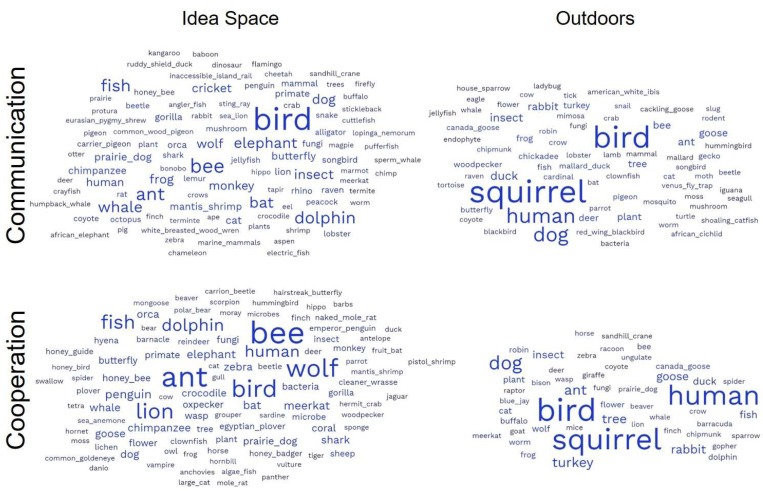
Word clouds generated from student responses and the list of represented taxonomic classes for the “Communication” and “Cooperation” assignments that were completed factorially with the outdoor and idea-space interventions. The idea-space intervention elicited a greater number of examples, a greater diversity of taxonomic classes, and examples that were more specific relative to the outdoor treatment. Raw frequency data can be found in [50] under an open access license.

**Table 1 biomimetics-08-00048-t001:** Focal problems. Background on how focal problem areas were discussed in class as context for the brainstorming activities.

Problem	Some of the Framing Used in the Course to Provide Context
Force	How do we build machines capable of generating extreme forces (e.g., to use in demolition) that do not harm the user? (e.g., vibration syndrome)
Anxiety	How do we address anxiety within the mental health crisis? How do we tamp down excess fear and chronic stress in modern life?
Grief	How can we help people to process the loss of a loved one? How do we move past grief that is holding us back from living?
Movement	How do we ensure we get enough exercise? How do we encourage ourselves to move around more?
Nutrition	How do we make healthy choices about what to eat? How do we pass on those salty snacks or sweet desserts?
Communication	How do we promote clear, effective, and honest communication? How do we ensure messages are understood in noisy and variable environments?
Cooperation	How do we encourage cooperative and nice interactions between people? How do we get people to cooperate with public health measures, safe driving practices, and other initiatives for the public good?

## Data Availability

All raw (anonymized) data and code used in the analysis are available on Mendeley [50].

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
