# Peer review of "Broadening the Taxonomic Breadth of Organisms in the Bio-Inspired Design Process"

_biomimetics, 2023, doi:10.3390/biomimetics8010048_

Round 1

Reviewer 1 Report

The work presented by Hund et al. investigated the effect of problems, individual expertise, interventions on the the taxonomic breadth of ideas and made an attempt to offer useful recommendations to increase the breadth of biological models generated in the bio-inspired design process. Generally, the topic discussed in this work is meaningful and the results are interesting for a broad readership. Before it can be accepted for publication in Biomimetics, there are some major concerns need to be addressed carefully.

1. As a research article, the title is too broad. Please modify to make it more focused.

2. The order numbers used in the Abstract section is odd. The abstract now is more like conclusion rather than abstract. Please reorganized this section.

3. The authors claimed that biological models such as geckos, butterflies, and spiders are overrepresented in biomimetic research (line 31) thus emphasized the value of other neglected species. As far as I know, these so-called overrepresented species like Morpho butterflies are usually featured with more typical and outstanding functions for specific bio-inspired design and application. I suggest the author to reorganize the relevant statements to avoid misunderstanding.

4. The biological model proposed by authors in this work seems different from what we are already familiar with in the natural science area. I suggest the clear definition of the biological model discussed here should be given at the very beginning.

5. The second (line 95) in the third (line 90) part makes me confused. Please check the similar issues in the whole manuscript to make your statements more smooth.

6. Violin plot is a commonly used plot to display data distribution and its probability density. The notes in the brackets (line 310 and 370) are needless.

7. The word cloud is indeed more visual to show frequency. But I suggest the detailed frequency data should also be included in the supplementary to make it more convincing.

8. The serial number of subtitles in 4. Discussion section is missing. Only 4.1. (line 498) can be found. Please carefully check the structure of the main text.

9. There is no Conclusion section in the manuscript? Please highlight your clear research conclusions as an independent section at the end.

Author Response

Reviewer 1. The work presented by Hund et al. investigated the effect of problems, individual expertise, interventions on the taxonomic breadth of ideas and made an attempt to offer useful recommendations to increase the breadth of biological models generated in the bio-inspired design process. Generally, the topic discussed in this work is meaningful and the results are interesting for a broad readership. Before it can be accepted for publication in Biomimetics, there are some major concerns need to be addressed carefully.

Response: Thank you for the overall positive assessment.

Reviewer 1. 1. As a research article, the title is too broad. Please modify to make it more focused.

Response: We have made the second part of the title more specific, so the title now reads “Exploring of biological models in the bio-inspired design process: how to broaden the taxonomic breadth of ideas in brainstorming.”

Reviewer 1. 2. The order numbers used in the “Abstract” section is odd. The abstract now is more like conclusion rather than “abstract”. Please reorganized this section.

Response: The numbers in the abstract follow the instructions to authors which reads “The abstract should be a total of about 200 words maximum. The abstract should be a single paragraph and should follow the style of structured abstracts, but without headings: 1) Background: Place the question addressed in a broad context and highlight the purpose of the study; 2) Methods: Describe briefly the main methods or treatments applied. Include any relevant preregistration numbers, and species and strains of any animals used. 3) Results: Summarize the article's main findings; and 4) Conclusion: Indicate the main conclusions or interpretations.”

To make the abstract less like a “conclusion” and more like an “abstract,” we have made some minor edits throughout the abstract, e.g., shifts to past tense in several places.

Reviewer 1. 3. The authors claimed that “biological models such as geckos, butterflies, and spiders are overrepresented in biomimetic research” (line 31) thus emphasized the value of other “neglected” species. As far as I know, these so-called “overrepresented” species like Morpho butterflies are usually featured with more typical and outstanding functions for specific bio-inspired design and application. I suggest the author to reorganize the relevant statements to avoid misunderstanding.

Response: We have added the phrase “While these species have unique traits that have inspired a number of applications” to clarify that these species are of course valuable to study, but expanding to consider other taxonomic groups is also valuable.

Reviewer 1. 4. The “biological model” proposed by authors in this work seems different from what we are already familiar with in the natural science area. I suggest the clear definition of the “biological model” discussed here should be given at the very beginning.

Response: This is a good point and one that we have discussed extensively. We have moved between terms such as species, system, and model and settled on model here given its use in the bio-inspired design field. To avoid confusion, we define it at first use in the introduction, as “The species that are the focus of bioinspiration, what we term “biological models,” (e.g., geckos, butterflies, spiders)…”

Reviewer 1. 5. The “second” (line 95) in the “third” (line 90) part makes me confused. Please check the similar issues in the whole manuscript to make your statements more smooth.

Response: Good point – we have changed this from “second” to “additionally” in reference to the second intervention. This part now reads “We additionally designed an intervention…”

Reviewer 1. 6. Violin plot is a commonly used plot to display data distribution and its probability density. The notes in the brackets (line 310 and 370) are needless.

Response: We added these explanations in response to initial reviews by students who had not seen violin plots before. We have removed the additional explanation.

Reviewer 1. 7. The word cloud is indeed more visual to show frequency. But I suggest the detailed frequency data should also be included in the supplementary to make it more convincing.

Response: The complete frequency data will be available in the full publically available dataset (e.g., the list of all student responses). We now include a reference to this in the legends for the word clouds as “Raw frequency data can be found in [Mendeley reference].”

Reviewer 1. 8. The serial number of subtitles in “4. Discussion” section is missing. Only “4.1.” (line 498) can be found. Please carefully check the structure of the main text.

Response: We have added an initial subheading of “findings” prior to the subheadings for “limitations” and “future research.”

Reviewer 1. 9. There is no “Conclusion” section in the manuscript? Please highlight your clear research conclusions as an independent section at the end.

Response: We have added a paragraph at the end of the discussion with conclusions where we summarize the main findings of the work: “Conclusions. In this research, we found strong support for the hypothesis that explicit consideration of evolutionary and ecological relationships can increase the taxonomic breadth of biological models generated during the brainstorming phase of a bio-inspired design process. An individual’s prior expertise with biological diversity contributed somewhat to their breadth of ideas, but exploring with online tools such as Wikipedia or the “tree of life” can push students into new idea spaces. We found no evidence that going outside, per se, increased the taxonomic breadth of ideas, but it is possible that combining this activity with more directed activities (e.g., “look under a rock”) would encourage students to move beyond the most obvious organisms encountered outdoors.

Reviewer 2 Report

This paper emphasis on the taxonomic diversity which plays a key role in bio-inspired design process for generating a variety of designs. The authors stress on the convergent evolution aspect where multiple organisms can solve same function in radically different ways.

The authors claim that divergent thinking will broaden the ideas of design space and is one of the many factors that lead to original, creative, and innovative ideas.

To verify, the authors have chosen a topic of Animal Behaviour and tested certain hypothesis namely,

a)      Prompt (focal problem often briefly discussed in class) vs Practice (through initial experience producing bio-inspired design)

b)      Individual Vs Team expertise on biological models

c)      Role of interventions (Online tools for searching for an inspiration and Outdoor activity for inspiration)

On 180 students through a problem-based brainstorming assignment (Online)

Results:

1)      Prompt topic has a significant effect on the number of taxonomic classes

2)      Practice did not appear to influence the measure of divergent thinking

3)      Individual expertise has no impact on number of examples nor student expertise relative to their group.

4)      Interaction with team members did not influence in addressing number of taxonomic classes

5)      Idea-space interventions generated more responses with greater number of taxonomic class representations and responses were more specific (e.g., species)

Discussion:

1)      The authors support the results by stating that there are limitations in generating more number of taxonomic classes due to lack of group dynamics, online course works which lack in quality of team interactions.

2)      The authors support the result of idea-intervention has generated more taxonomic classes by stating that online tools would explicitly push students towards new biological systems and enhanced taxonomic diversity.

Recommendations & Conclusions

1)      Partnership with biologist with complementary expertise would be extremely beneficial.

2)      Missing search tool to promote taxonomic breadth and making the analogy between design problem and biological model.

3)      Promote collaboration between biologists, engineers, and scientists for bio-inspired design.

Author Response

Reviewer 2. This paper emphasis on the taxonomic diversity which plays a key role in bio-inspired design process for generating a variety of designs. The authors stress on the convergent evolution aspect where multiple organisms can solve same function in radically different ways.

The authors claim that divergent thinking will broaden the ideas of design space and is one of the many factors that lead to original, creative, and innovative ideas.

To verify, the authors have chosen a topic of Animal Behaviour and tested certain hypothesis namely,

  1. a)Prompt (focal problem often briefly discussed in class) vs Practice (through initial experience producing bio-inspired design)
  2. b)Individual Vs Team expertise on biological models
  3. c)Role of interventions (Online tools for searching for an inspiration and Outdoor activity for inspiration)

On 180 students through a problem-based brainstorming assignment (Online)

Results:

1)      Prompt topic has a significant effect on the number of taxonomic classes

2)      Practice did not appear to influence the measure of divergent thinking

3)      Individual expertise has no impact on number of examples nor student expertise relative to their group.

4)      Interaction with team members did not influence in addressing number of taxonomic classes

5)      Idea-space interventions generated more responses with greater number of taxonomic class representations and responses were more specific (e.g., species)

Discussion:

1)      The authors support the results by stating that there are limitations in generating more number of taxonomic classes due to lack of group dynamics, online course works which lack in quality of team interactions.

2)      The authors support the result of idea-intervention has generated more taxonomic classes by stating that online tools would explicitly push students towards new biological systems and enhanced taxonomic diversity.

Recommendations & Conclusions

1)      Partnership with biologist with complementary expertise would be extremely beneficial.

2)      Missing search tool to promote taxonomic breadth and making the analogy between design problem and biological model.

3)      Promote collaboration between biologists, engineers, and scientists for bio-inspired design.

Response: We appreciate the reviewer’s summary of our motivations and findings. We do not see any critiques or requested changes within this review, so no resulting changes have been made.

Round 2

Reviewer 1 Report

The authors have made appropriate modifications in the revision version. It can be accepted for publication after minor revision, such as delete the full stop in the title. Please also check and modify the similar format and grammar issues in the whole manuscript.

Author Response

Reviewer 1: The authors have made appropriate modifications in the revision version. It can be accepted for publication after minor revision, such as delete the full stop in the title. Please also check and modify the similar format and grammar issues in the whole manuscript.

Response: We have modified the title to one sentence.

We have also done a final read-through to correct remaining grammar issues. We added one reference in response to input from the SI editor.